# Mesoporous Dual-Semiconductor ZnS/CdS Nanocomposites as Efficient Visible Light Photocatalysts for Hydrogen Generation

**DOI:** 10.3390/nano13172426

**Published:** 2023-08-26

**Authors:** Ioannis Vamvasakis, Evangelos K. Andreou, Gerasimos S. Armatas

**Affiliations:** Department of Materials Science and Technology, University of Crete, 70013 Heraklion, Greece; vaggelisandr@gmail.com (E.K.A.); garmatas@materials.uoc.gr (G.S.A.)

**Keywords:** cadmium sulfide, zinc sulfide, metal chalcogenides, nanoparticles, mesoporous materials, nanocomposites, ZnS/CdS heterojunctions, water splitting, photocatalytic hydrogen production

## Abstract

The development of functional catalysts for the photogeneration of hydrogen (H_2_) via water-splitting is crucial in the pursuit of sustainable energy solutions. To that end, metal-sulfide semiconductors, such as CdS and ZnS, can play a significant role in the process due to their interesting optoelectronic and catalytic properties. However, inefficient charge-carrier dissociation and poor photochemical stability remain significant limitations to photocatalytic efficiency. Herein, dual-semiconductor nanocomposites of ZnS/CdS nanocrystal assemblies (NCAs) are developed as efficient visible light photocatalysts for H_2_ generation. The resultant materials, synthesized via a polymer-templated self-polymerization method, comprise a unique combination of ~5–7 nm-sized metal-sulfide nanoparticles that are interlinked to form a 3D open-pore structure with large internal surface area (up to 285 m^2^ g^−1^) and uniform pores (circa 6–7 nm). By adjusting the ratio of constituent nanoparticles, the optimized ZnS/CdS catalyst with 50 wt.% ZnS content demonstrates a remarkable stability and visible light H_2_-evolution activity (~29 mmol g^−1^ h^−1^ mass activity) with an apparent quantum yield (AQY) of 60% at 420 nm. Photocatalytic evaluation experiments combined with electrochemical and spectroscopic studies suggest that the superior photocatalytic performance of these materials stems from the accessible 3D open-pore structure and the efficient defect-mediated charge transfer mechanism at the ZnS/CdS nanointerfaces. Overall, this work provides a new perspective for designing functional and stable photocatalytic materials for sustainable H_2_ production.

## 1. Introduction

The exponential growth of urbanization and population has led to a significant surge in the consumption of resources, giving rise to global challenges such as energy scarcity and environmental deterioration. At present, over 80% of the world’s energy supply still comes from the combustion of non-renewable fossil fuels, such as oil, coal and natural gas, leading to the release of CO_2_ and other harmful gases which contribute significantly to climate change and generate adverse effects on the environment and human health [1,2]. Addressing this issue is undeniably a complex challenge, as it is imperative to establish eco-friendly and renewable technological solutions to foster a more sustainable future. In this context, photocatalysis has gained significant importance in the development of alternative energy sources, particularly in the areas of green energy conversion and the production of clean fuels. Among the various photocatalytic processes, the production of hydrogen (H_2_) via semiconductor-based photochemical water-splitting has been regarded as one of the “holy grails” in the future energy portfolio. This is because water is a renewable resource, and H_2_ possesses the advantages of having a high energy content (~122 kJ g^−1^) and being a clean (zero-carbon emission) energy carrier [3,4]. Key to this approach is the development of new photocatalytic materials that should combine cost-efficiency (i.e., be inexpensive and synthesized via “green” and economically viable processes) with high photochemical performance and stability [5]. To that end, numerous photo-active metal oxide- and chalcogenide-based materials have been extensively studied and reviewed as photocatalysts for H_2_ generation via water-splitting [6,7,8]. Among them, II-VI metal-sulfide semiconductors, such as cadmium sulfide (CdS) and zinc sulfide (ZnS), have attracted enormous attention due to their interesting electronic and catalytic properties. In particular, ZnS has demonstrated high activity for H_2_ generation due to its high photon-to-electron conversion efficiency and largely negative conduction band potential, well above the reduction potential of protons (−0.41 V vs. NHE at pH 7) [9,10]. However, the wide band-gap energy of ZnS (~3.6 eV) imposes a significant limitation on the efficient utilization of solar energy (ZnS absorbs only UV light, which constitutes about 4% of the solar spectrum), thereby restricting its practical application. On the other hand, with a lower band-gap energy of ~2.3–2.5 eV and appropriate band-edge positions for water-splitting, CdS plays a significant role in the field of photocatalytic H_2_ production owing to its visible light response and high electron mobility [11,12,13,14]. However, its hydrogen evolution activity is mostly hindered by the rapid recombination of photoexcited electron-hole pairs and photocorrosion, in which the photogenerated holes oxidize the S^2−^ lattice sites and cause anodic decomposition of CdS [15]. Recently, several studies have reported that coupling CdS with ZnS to create dual-semiconductor composites [16,17,18,19,20] or core-shell heterostructures [21,22,23,24,25,26,27] remarkably enhances visible light catalytic activity for H_2_ generation. This approach partially addresses some of the limitations associated with ZnS, such as its unsuitable band gap for visible light absorption, as well as the carrier recombination losses and photocorrosion in CdS, by promoting spatial separation of charges across the ZnS/CdS interface [23,25]. However, the photocatalytic efficiency of many of these materials remains rather limited, primarily due to the inadequate synergistic effect resulting from the unfavorable distribution and inefficient contact between the ZnS and CdS components, as well as the limited exposure of active sites. In this regard, the design and synthesis of ZnS/CdS composites with open-pore structure and uniform blend of nanosized constituents present a promising strategy to improve the catalytic performance.

We have recently developed a new polymer-templated self-polymerization method that allows for the assembly of small-sized (circa 4–7 nm) metal sulfide nanocrystals (NCs) into 3D mesoporous architectures [28]. These materials combine the advantages of high internal surface area that provides a large exposure of active sites, and short diffusion pathways for charge carriers to reach the surface, thereby facilitating high photocatalytic reaction rates. As a result, significantly enhanced photocatalytic H_2_ generation activities have been demonstrated for various mesoporous CdS-based NC ensembles, far surpassing those observed for the respective individual nanoparticles and bulk microcrystalline analogues [29,30,31,32]. In this work, we report the synthesis of new 3D mesoporous frameworks consisting of interconnected ZnS and CdS NCs and investigate their visible light photocatalytic performance for H_2_ generation. The dual-semiconductor nanocomposite structures were synthesized via the slow oxidative coupling of thiol-capped CdS and ZnS NCs in the presence of an amphiphilic polyoxyethylene-*block*-cetyl ether (Brij-58) block copolymer acting as a pore-forming template. Comprehensive structural and morphological characterizations showed that the resulting ZnS/CdS nanocrystal assemblies (NCAs) are composed of connected small-sized (circa 5–7 nm) CdS and ZnS nanoparticles forming a 3D framework with a varying ZnS composition (ranging from 10 to 90 wt.%) that is perforated by uniform mesopores. Additionally, by employing a combination of electrochemical and spectroscopic techniques, we gain valuable insights into the band structure and charge transfer dynamics within this catalytic system. Finally, based on the experimental results, we discuss a plausible quasi-type-II charge transfer mechanism that elucidates the efficient spatial separation of photogenerated charge carriers and accounts for the improved photocatalytic H_2_-production activity observed in the ZnS/CdS heterojunctions.

## 2. Materials and Methods

### 2.1. Materials

Ammonia solution (NH_3_·H_2_O, 25–28%), sodium sulfide nonahydrate (Na_2_S·9H_2_O), sodium sulfite anhydrous (Na_2_SO_3_), cadmium chloride (CdCl_2_) and zinc chloride (ZnCl_2_) were acquired from Sigma-Aldrich (Steinheim, Germany). 3-Mercaptopropionic acid (3-MPA, 99%) was obtained from Alfa Aesar, and block copolymer (Brij 58, Mn ~1124 g mol^−1^) from Aldrich Chemical Co. Absolute ethanol and isopropanol (99%) were purchased from Fischer Scientific Company (Hampton, NH, USA). All chemical reagents were used as received without any further purification, and double deionized (DI) water was used in all procedures.

### 2.2. Synthesis of Colloidal CdS and ZnS NCs

The colloidal CdS and ZnS NCs were prepared via a wet-chemical procedure using CdCl_2_ and ZnCl_2_ as metal sources and Na_2_S·9H_2_O as a sulfur reagent. 3-mercaptopropionic acid (3-MPA) was used as thiol-capping agent to stabilize the NCs. For the synthesis of CdS and ZnS NCs, a typical metal:3-MPA:sulfur ratio of 1:2:1 was employed. In particular, 10 mmol of CdCl_2_ (or ZnCl_2_) and Na_2_S·9H_2_O were dissolved separately in 40 and 20 mL of DI water, respectively. Then, 20 mmol of 3-MPA were added to the CdCl_2_ (or ZnCl_2_) solution and left under stirring for 30 min at room temperature (RT). The pH was then adjusted to ~10 using ammonia solution, followed by the slow addition of the Na_2_S solution under vigorous stirring. The resultant mixtures were stirred for an additional 1 h at RT, forming clear colloidal dispersions of thiol-capped CdS (or ZnS) NCs. Finally, the thiol-capped CdS (or ZnS) NCs were isolated by precipitation with isopropanol, followed by centrifugation and drying at 40 °C for ~24 h.

### 2.3. Synthesis of Mesoporous ZnS, CdS and ZnS/CdS NC Assemblies (NCAs)

The mesoporous CdS and ZnS NC assemblies (NCAs) were prepared according to our previously reported method [28], with slight modifications. In brief, 300 mg of thiol-capped CdS (or ZnS) NCs were dispersed in 5 mL of DI water containing ~0.1 mL ammonia solution, forming a clear colloidal suspension (sol). Afterwards, 300 mg of Brij 58 was added, and the mixture was left under vigorous stirring for 1 h at RT, resulting in a stable colloidal dispersion. Next, 1 mL of 3% (*v*/*v*) aqueous H_2_O_2_ solution was added and, after stirring for an additional 5 min, the resultant solution was left under static conditions until gelation was observed (after ~1 h). After gelation, the mixture was placed in an oven at 40 °C and left for ~3 days to evaporate the solvent. The template removal was achieved via successive washing cycles, which included the soaking and stirring of the dry gel product in 20 mL ethanol for 30 min, followed by vacuum filtration, and repeating this process three more times with DI water. The final mesoporous CdS and ZnS NCAs were obtained after drying the respective washed samples at 60 °C for ~12 h.

The dual-semiconductor ZnS/CdS NCAs were prepared following the exact same procedure but by employing appropriate mixtures of thiol-capped CdS and ZnS NCs to achieve the desired mass loadings (wt.%) of ZnS in the final heterostructures. For instance, to maintain the total quantity of metal-sulfide NCs during the reaction fixed at 300 mg, a mixture of 150 mg CdS and 150 mg ZnS NCs was used to prepare the mesoporous ZnS/CdS nanocomposite with 50 wt.% ZnS content.

### 2.4. Synthesis of Bulk ZnS/CdS

For comparison purposes, a ZnS/CdS bulk heterostructure with 50 wt.% ZnS content was prepared by physically mixing two separate 10 mL aqueous dispersions containing the appropriate amounts of pre-formed CdS and ZnS microparticles. The CdS and ZnS microparticles were synthesized via direct precipitation of CdCl_2_ and ZnCl_2_ with equimolar amounts of Na_2_S in water, respectively. After mixing, the resulting ZnS/CdS suspension was stirred for about 3 h at RT, followed by centrifugation and drying at 100 °C overnight. 

### 2.5. Physicochemical Characterization

Energy-dispersive X-ray spectroscopy (EDS) and field-emission electron microscopy (FE-SEM) were conducted using a JEOL JSM-IT700HR microscope equipped with a JED-2300 detector (JEOL Ltd., Tokyo, Japan). EDS data acquisition was carried out at least ten times (at different locations) for each sample using an accelerating voltage of 20 kV and a 60 s accumulation time. X-ray diffraction (XRD) patterns were obtained with a PANanalytical X’ pert Pro MPD diffractometer (Malvern PANalytical Ltd., Almelo, The Netherlands) equipped with a Cu Kα (λ = 1.5418 Å) radiation source in Bragg–Brentano geometry (45 kV and 40 mA). Transmission electron microscopy (TEM) imaging was performed with a JEOL JEM-2100 microscope (JEOL Ltd., Tokyo, Japan) equipped with a LaB_6_ filament, using an acceleration voltage of 200 kV. TEM samples were prepared by depositing few drops of an ethanolic dispersion containing the respective materials onto a perforated carbon-coated Cu grid. X-ray photoelectron spectroscopy (XPS) measurements were taken with a SPECs spectrometer (SPECS Surface Nanon Analysis, Berlin, Germany), using a Phoibos 100 1D-DLD electron analyzer and Al Kα monochromatic radiation (1486.6 eV). In all cases, the binding energies were corrected with respect to the C1 signal of adventitious carbon at 284.8 eV. Thermogravimetric analysis (TGA) was performed with a Perkin-Elmer Diamond system under a nitrogen flow of 100 mL min^−1^ with a heating rate of 5 °C min^−1^. N_2_ physisorption (adsorption–desorption) isotherms were recorded at −196 °C with a Quantachrome NOVA 3200e sorption analyzer (Quantachrome Instruments, Boynton Beach, FL, USA). Before measurement, the samples were degassed at 80 °C under low pressure (<10^−5^ Torr) for 12 h. The specific surface areas were estimated from the adsorption data in the relative pressure (P/P_0_) range of 0.04–0.22 using the Brunauer–Emmett–Teller (BET) method. The total pore volumes were determined from the amount of adsorbed N_2_ at P/P_0_ = 0.98. The pore size distributions were derived from the adsorption branch of the isotherms using the nonlocal density functional theory (NLDFT) fitting model. A Shimazu UV-2600 spectrophotometer (Shimadzu Co., Kyoto, Japan) was employed to acquire the UV-vis/near-IR diffuse reflectance spectra, using BaSO_4_ powder as the 100% reflectance reference. The conversion of the diffuse reflectance data to absorbance (α/S) was attained using the Kubelka–Munk function: α/S = (1 − R)^2^/(2R), where R represents the measured reflectance, while α and S denote the absorption and scattering coefficients, respectively. The energy band gaps (E_g_) of the samples were determined from Tauc plots for direct allowed transitions, i.e., the plot of (α*hv*)^2^ vs. *hv*, where α is the absorption coefficient and *hv* is the energy of incident photons. Photoluminescence (PL) and time-resolved photoluminescence (TR-PL) spectra were acquired at RT using an Edinburgh FS5 spectrofluorometer equipped with a 150 W CW ozone-free xenon arc lamp and a 375 nm EPL picosecond pulsed diode laser (Edinburgh Instruments Ltd., Livingston, UK).

### 2.6. Photocatalytic Study

The photocatalytic hydrogen evolution experiments were performed in an airtight Pyrex glass reactor, which was placed in a water-bath cooling system to ensure a constant reaction temperature of 20 ± 2 °C. In a typical experiment, 20 mg of catalyst was dispersed in a 20 mL aqueous electrolyte solution containing 0.35 M Na_2_S and 0.25 M Na_2_SO_3_ (except when otherwise noted). Prior to measurement, the reaction mixture was thoroughly deaerated via purging with argon (Ar) for at least 30 min, and then irradiated with visible light using a 300 W Xe lamp (Variac Cermax) equipped with a UV cut-off filter (Asahi Techno Glass, λ ≥ 420 nm). The amount of evolved H_2_ was determined by extracting 100 μL gas samples from the headspace of the reactor at specific time intervals (using a gastight syringe) and analyzing them on a Shimadzu GC-2014 gas chromatograph (using Ar as a carrier gas) equipped with a thermal conductivity detector (TCD).

Photocatalytic stability tests were carried out under visible light irradiation by performing consecutive photocatalytic H_2_-evolution runs over specific time periods (cycles). After each cycle, the catalyst was collected via centrifugation and then redispersed in a fresh Na_2_S/Na_2_SO_3_ solution. Additionally, prior to each catalytic run, the reaction mixture was thoroughly deaerated with Ar gas for at least 30 min until no H_2_, O_2_ or N_2_ gases were detected via gas chromatography (GC).

The apparent quantum yield (AQY) was estimated by analyzing the average rate of hydrogen evolution obtained under monochromatic LED light (λ = 420 ± 10 nm) irradiation according to the following equation: (1)$$\mathrm{AQY}=\frac{2\times N_{H2}}{N_{\mathrm{hv}}}\times 100\%$$
where *N*_H2_ is the flux of evolved H_2_ molecules (i.e., the number of evolved H_2_ molecules per unit time) and *N*_hv_ is the flux of incident photons at λ = 420 nm (i.e., number of photons per unit time). The average intensity of incident light was determined using a StarLite power meter equipped with a FL400A-BB-50 fan-cooled thermal sensor (Ophir Optronics Ltd., Jerusalem, Israel) and was found to be 11.7 mW cm^−2^. This corresponds to a flux of incident photons of approximately 7.6 × 10^16^ s^−1^.

### 2.7. Electrochemical Measurements

Electrochemical Mott–Schottky and Nyquist impedance measurements were performed using a VersaSTAT 4 single-channel potentiostat–galvanostat electrochemical workstation (Princeton Applied Research, Princeton, NJ, USA). All experiments were conducted in a 0.5 M Na_2_SO_4_ electrolyte (pH = 7), using a custom-made three-electrode electrochemical cell consisting of a sample-coated fluorine-doped tin oxide (FTO, 10 Ω sq^–1^) working electrode, a Ag/AgCl (saturated KCl) reference electrode, and a graphite-rod counter-electrode. For the preparation of the working electrodes, 10 mg of each sample was first dispersed in 1 mL of absolute ethanol using ultrasonication for 30 min, followed by vigorous stirring for 1 h at RT to form uniform suspensions. Then, each sample suspension was gradually drop-cast onto the conductive surface of FTO glass substrates (working surface area of 1 cm^2^) under mild heating conditions (50–60 °C) to produce thick films in order to ensure that the width of the depletion layer did not exceed the thickness of the semiconductor layer [33].

To obtain the Mott–Schottky plots, the interfacial capacitance (C) of the samples was measured at a frequency of 1 kHz using a 10 mV AC voltage amplitude. All the measured potentials were converted to the reversible hydrogen electrode (RHE) scale at pH = 7 using the equation:E_RHE_ = E_Ag/AgCl_ + 0.197 (2)
where E_RHE_ is the potential in the RHE scale and E_Ag/AgCl_ is the measured potential in the Ag/AgCl scale. 

The flat-band potential (E_FB_) and donor density (N_D_) of the samples were estimated from the linear portion of the Mott–Schottky curves according to the Mott–Schottky equation [33,34]:(3)$$\frac{1}{C^{2}}=\frac{2}{{\varepsilon \varepsilon }_{o}A^{2}e_{o}N_{D}}\;\left(E-E_{\mathrm{FB}}-\frac{k_{B}T}{e_{o}}\right)=\frac{2\left(E-E_{\mathrm{FB}}\right)}{{\varepsilon \varepsilon }_{o}A^{2}e_{o}N_{D}}$$
where C is the measured interfacial capacitance, E is the applied potential, E_FB_ is the flat band potential, A is the working area of the electrode (1 cm^2^), ε is the relative dielectric constant of the material (8.9) [35,36], ε_o_ is the dielectric permittivity under vacuum (8.8542 × 10^–14^ F cm^–1^), e_o_ is the elementary charge (1.602 × 10^–19^ C), k_B_ is the Boltzmann constant, and T is the temperature. Note that, under normal conditions, the term “*k_B_T/e_o_*” is usually negligible [37] and the Mott–Schottky relation can be simplified as shown in Equation (3). Thus, the corresponding E_FB_ values were determined from the linearly extrapolated intercepts with the *x*-axis (1/C^2^ = 0), while N_D_ values were calculated from the reciprocal of the slopes of the Mott–Schottky plots and using Equation (3).

The Nyquist impedance measurements were conducted in 0.5 M Na_2_SO_4_ electrolyte (pH = 7) using an applied DC bias of −1.3 V (vs. Ag/AgCl, saturated KCl) and the data were recorded over a frequency range from 1 Hz to 100 KHz. ZView software was used to fit the experimental data to a simplified Randles circuit model, consisted of an electrolyte resistance (R_s_), a charge transfer resistance (R_ct_) and a double-layer capacitance (C_dl_).

## 3. Results and Discussion

### 3.1. Materials Synthesis and Characterization

The synthesis of mesoporous ZnS/CdS nanocomposites was accomplished via an evaporation-induced self-assembly (EISA) sol-gel method, in which ZnS and CdS colloidal NCs self-organized into 3D mesoporous gel-like networks with the aid of block copolymer (POE_20_-*b*-C_16_; Brij 58) micelles, as illustrated in Figure 1a. To synthesize the CdS and ZnS colloidal NCs, a thiol-terminated ligand (3-mercaptopropionic acid) was employed, which can exert control over the nanoparticle growth by binding to the NC surface via the thiol group and can facilitate enthalpic interactions with the polar POE fragments of the polymer template via the propionic acid end-group [28]. The pre-formed NCs were subsequently utilized as building-block units that co-assemble with the polymer micelles under solvent evaporation conditions to create hybrid NC/polymer mesostructures. Key to this step is the addition of a small amount of dilute H_2_O_2_ solution (3% *v*/*v*) to the NC/polymer dispersion (as outlined in the experimental section), which instigates the gradual oxidation and stripping of the thiolate ligands from the NCs’ surface and prompts the oxidative polymerization of metal-sulfide NCs into a 3D network (on the surface of the polymer template) via interparticle disulfide or polysulfide linking bonds [38,39,40]. It is noteworthy that the oxidant concentration and the related oxidation rates of the stabilizing thiolate capping groups constitute critical parameters that play a pivotal role in the controlled aggregative assembly of colloidal NCs and the preparation of robust gel-like porous frameworks. This observation builds upon earlier research into the formation of metal chalcogenide gels and aerogels, where it was demonstrated that the addition of a small quantity of oxidant (e.g., H_2_O_2_) below a specific threshold value (minimum ratio of oxidant/thiolate, X_min_) did not lead to sufficient removal of thiolate capping groups to initiate the gelation–polymerization of NCs [41]. Above X_min_, efficient oxidation of surface thiolate ligands was achieved, resulting in interparticle linking and the assembly of stable gel networks, which can only be disassembled under (electro)chemical reductive environments [38,40]. Conversely, the utilization of a high amount of oxidant (~5X_min_) triggered a rapid loss of surface thiolate groups, resulting in uncontrolled NC agglomeration and the precipitation of densely packed aggregates [41,42,43]. Therefore, by employing the appropriate amount of H_2_O_2_, the creation of an interconnected NC-network provided sufficient structural stability to enable the effective removal of the organic template from the hybrid mesostructures via a gentle post-preparative treatment with ethanol and water (as detailed in the experimental section), thus yielding mesoporous ZnS/CdS NCAs. Consequently, via this procedure and the concentration of the colloidal CdS and ZnS nano-building blocks were appropriately adjusted, a series of mesoporous ZnS/CdS nanocomposites with varying mass loadings (wt.%) of ZnS were prepared, designated as x-ZnS/CdS (x = 10, 30, 50, 70 and 90 wt.% ZnS). Additionally, single-component mesoporous CdS and ZnS NCAs were prepared using the same synthetic process and extensively studied for comparison purposes. The chemical composition of the resultant materials was determined via energy-dispersive X-ray spectroscopy (EDS), and the obtained results are summarized in Appendix A. For the CdS NCAs, EDS analysis revealed the presence of Cd and S elements with a Cd:S atomic ratio very close to the 1:1 stoichiometry of CdS. In contrast, the corresponding Zn:S atomic ratio in the ZnS NCAs sample was found to be ~1:0.86, implying a small sulfur deficiency (approximately 14%) in the lattice of constituent ZnS NCs. The EDS results of the ZnS/CdS nanocomposites indicated the presence of Cd and Zn atoms at varying Cd/Zn ratios, based on which the ZnS loading amounts were estimated and found to be very close to the nominal compositions of 10–90 wt.% (see Appendix A).

The crystallinity of the as-prepared materials was characterized via powder X-ray diffraction (XRD). As seen in Figure 1b, the mesoporous CdS and ZnS NCAs exhibit three broad diffraction peaks at the scattering angle (2θ) range of 20−60°, which can be assigned to the (111), (220), and (311) reflections of the cubic (hawleyite) structure of CdS (JCPDS no. 42–1411) and the sphalerite (cubic) crystal phase of ZnS (JCPDS no. 05–0566). The broadening of the XRD peaks reflects the presence of very small nanocrystallites with an average domain size of ~2.8 nm for CdS and ~3.1 nm for ZnS, as determined via Scherrer analysis of the (111) diffraction peak [44]. The ZnS/CdS NCAs showed XRD patterns that resemble a combination of those of CdS and ZnS, where the corresponding peaks of ZnS gradually increase in intensity with increasing the ZnS content. This suggests the formation of dual-component ZnS/CdS composites rather than solid solution structures via incorporation of ZnS nanoclusters into the CdS matrix.

The morphology of the ZnS/CdS nanocomposite with 50 wt.% ZnS content (50-ZnS/CdS), which is the most active catalyst in this study, was assessed via field emission scanning electron microscopy (FE-SEM) and transmission electron microscopy (TEM). The representative FE-SEM image in Figure 2a reveals an agglomerated morphology comprising small-sized nanoparticles within the range of 5 to 10 nm. In Figure 2b, the SEM-EDS elemental mapping shows a uniform dispersion of Cd, Zn and S elements, indicating that the ZnS and CdS nanoparticles are evenly distributed throughout the assembled structure. The TEM image in Figure 2c further reveals an open-pore structure configuration, while the high magnification image (Figure 2c inset) also demonstrates that the pore walls consist of approximately 5–6 nm-sized CdS and ZnS nanoparticles that are in intimate contact, which is advantageous for promoting efficient interfacial charge transfer and separation. Furthermore, the high-resolution TEM (HRTEM) image in Figure 2d displays distinct lattice fringes with interplanar spacings of 3.4 Å and 3.1 Å, corresponding to the (111) crystal planes of the cubic CdS and ZnS, respectively, in agreement with the XRD results. Taken together, the above findings attest to the successful synthesis of porous ZnS/CdS nanocomposites, comprising small-sized ZnS and CdS nanoparticles that are linked together to form a robust 3D network. 

The chemical state of elements in the 50-ZnS/CdS material was analyzed via X-ray photoelectron spectroscopy (XPS). The XPS spectra of pure CdS and ZnS NCAs were also obtained as reference. Typical survey spectra are shown in Appendix A, while high-resolution spectra of the Cd 3d, Zn 2p and S 2p regions are shown in Figure 3. The Cd 3d spectra show strong signals at binding energies of 412.2 and 405.5 eV for CdS NCAs and 411.9 and 405.2 eV for the 50-ZnS/CdS nanocomposite (Figure 3a), which are respectively assigned to the Cd 3d_3/2_ and Cd 3d_5/2_ core-levels of CdS [45,46]. The Zn 2p XPS spectrum of ZnS NCAs in Figure 3b presents two peaks at 1045.0 and 1022.0 eV that correspond to the Zn 2p_1/2_ and Zn 2p_3/2_ core-levels of ZnS, respectively [47,48]. The respective Zn 2p binding energies of 50-ZnS/CdS are located at 1045.3 and 1022.3 eV. Furthermore, the high-resolution S 2p spectra (Figure 3c) of all the examined samples display a prominent peak at 161.8–162.2 eV that is characteristic of the S^2−^ valence state in metal sulfides [45,46,47,48]. The additional small S 2p XPS peaks observed at higher binding energies (168.5–169 eV) can be attributed to some oxidized sulfur species (SO_x_^2−^) formed on the particle surface during the synthesis process and/or air handling of the samples [45,49]. Noticeably, the binding energies of Cd 3d in the 50-ZnS/CdS nanocomposite are shifted by 0.3 eV towards lower energy in comparison with those of pure CdS NCAs, while the respective Zn 2p binding energies exhibit an upshift of 0.3 eV relative to the pure ZnS NCAs. These particular shift patterns in binding energies can be attributed to the altered chemical environment at the ZnS/CdS interface resulting from an electron transfer from ZnS to CdS upon hybridization, indicating the strong electronic interaction between the CdS and ZnS components.

The ideal outcome of the employed polymer-templated oxidative polymerization process would be the creation of materials with a porous interconnected network of NCs devoid of any organic molecules. In practice, however, it is common to observe a certain quantity of residual organics (<8 wt.%) on the surface of the NC-assembled structure, even after thorough washing procedures to remove the polymer template [28]. The presence of residual organics can influence factors such as wetting properties (hydrophobicity) and accessible surface area, and hence, the photocatalytic activity of the resulting porous structures [39]. In light of this, we assessed the efficiency of organic matter removal from the fabricated mesoporous NCA structures via thermogravimetric analysis (TGA). An illustrative example of TGA characterization performed on the mesoporous 50-ZnS/CdS nanocomposite (which is the best-performing catalyst of this work) is presented in Appendix A. In the TGA profile, the initial weight loss (~2.4%) observed in the temperature range of 50–200 °C can be attributed to the removal of adsorbed water, indicating the hydrophilic nature of the surface. Furthermore, the analysis revealed that approximately 7.4 wt.% of organic residue remains within the assembled structure, as indicated by the weight loss in the temperature range of 200–400 °C. This observation aligns with our earlier findings regarding a CdS NCA structure [28]. Nevertheless, as we show below (in N_2_ physisorption and photocatalytic H_2_ production experiments), this small percentage of residual organic content does not exert any significant constraints on pore accessibility or adversely impact the catalytic surface properties of the synthesized mesoporous nanocomposites. The porosity of the prepared materials was examined via N_2_ physisorption measurements conducted at −196 °C. Figure 4a presents a comparison of the N_2_ adsorption–desorption isotherms and the corresponding pore size distribution plots for the mesoporous CdS and ZnS NCAs, along with those of the 50-ZnS/CdS nanocomposite. The corresponding plots for the other ZnS/CdS NCAs (with 10, 30, 70 and 90 wt.% ZnS content) are provided in Appendix A. All the isotherms feature typical type-IV curves with H2-type hysteresis loop, which are characteristic of porous solids with interconnected pores [50]. Additionally, all the N_2_ adsorption isotherms exhibit a small but distinguishable capillary condensation step in the relative pressure (P/P_o_) range of ~0.4–0.6, suggesting N_2_ condensation in narrow-sized mesopores [51]. The Brunauer−Emmett−Teller (BET) surface area (S_BET_) and total pore volume (V_p_) of the mesoporous ZnS/CdS NCAs were measured to be in the range of 243–285 m^2^ g^−1^ and 0.29–0.32 cm^3^ g^−1^, respectively, which fall between those of pure CdS (S_BET_ = 234 m^2^ g^−1^, V_p_ = 0.28 cm^3^ g^−1^) and ZnS (S_BET_ = 334 m^2^ g^−1^, V_p_ = 0.33 cm^3^ g^−1^) NCAs. As shown in Table 1, the S_BET_ and V_p_ of the ZnS/CdS nanocomposites progressively increased as the ZnS content increased from 10 to 90 wt.%. This trend is likely attributed to the relatively lower density of ZnS (4.09 g cm^−3^) compared to that of CdS (4.29 g cm^−3^). Moreover, the pore width of the as-prepared mesoporous materials was estimated by applying the non-local density functional theory (NLDFT) fitting model to the adsorption data [52]. The NLDFT analysis indicated that all the samples have quite narrow pore-size distributions (insets of Figure 4a and Appendix A) with an average pore size (D_p_) of ~6.8 and ~5.8 nm for the CdS and ZnS NCAs, respectively, and in the range of ~5.8–6.3 nm for the ZnS/CdS nanocomposites (see Table 1). 

The effect of ZnS loading on the electronic structure and optical absorption properties of the synthesized ZnS/CdS nanocomposites was examined via UV-visible/near-IR diffuse reflectance spectroscopy (UV-vis/NIR DRS). The optical absorption spectra (Figure 4b) of the reference CdS and ZnS NCAs, obtained by transforming the diffuse reflection data using the Kubelka–Munk function, indicate sharp absorption onsets at 495 and 322 nm that correspond to the interband (i.e., from valence band to conduction band) electron transitions in CdS and ZnS, respectively. Interestingly, the spectra of the heterostructured ZnS/CdS samples display two well-defined absorption onsets, one in the visible light range (400–500 nm) and the other in the UV range (260–360 nm). These distinct absorption bands correspond to the respective light absorption from the constituent CdS and ZnS NCs, thereby signifying the dual-semiconductor nature of the as-synthesized ZnS/CdS NCAs. Additionally, a noticeable trend is observed in the absorption edges of the CdS and ZnS components of the ZnS/CdS nanocomposites. Specifically, the absorption edge of CdS shows a consistent blue-shift from 488 nm to 475 nm as the ZnS content increases from 10 to 90 wt.%. Conversely, the absorption edge of ZnS exhibits a red-shift from 327 nm to around 355 nm with the increase in ZnS content. This trend is related to the strong electron interactions between the electronic states in CdS and ZnS NCs, demonstrating their good interfacial connectivity in the assembled frameworks, in line with the results from TEM and XPS analyses. The energy band gaps (E_g_) of the CdS, ZnS and ZnS/CdS mesoporous structures were determined from the corresponding Tauc plots for direct band gap semiconductors (i.e., (αhν)^2^ vs. hν plots, derived from the UV−vis/NIR DRS spectra), as shown in Figure 4c. The E_g_ values of pure CdS and ZnS NCAs were estimated to be 2.51 ± 0.01 eV and 3.85 ± 0.02 eV, respectively (see Table 1). Compared to the band gap energies of bulk CdS (2.3–2.4 eV) and ZnS (3.6–3.7 eV), both mesoporous samples exhibit a small but resolved blue-shift in E_g_, which is indicative of a quantum confinement effect in the constituent CdS and ZnS nanoparticles due to their very small size (circa 5–6 nm according to the TEM analysis). Meanwhile, the strong interfacial interactions between CdS and ZnS NCs are also reflected in the band gaps of the ZnS/CdS nanocomposites, where their E_g_ values systematically vary from 2.54 eV to 2.61 eV with increasing the ZnS content from 10 to 90 wt.% (Table 1). Note that the referred E_g_ values for the ZnS/CdS nanocomposites were estimated from the respective absorption bands in the visible light region of the Tauc plots, as shown in the inset of Figure 4c. Despite the widening of the band gap, however, all the prepared mesoporous ZnS/CdS NCAs exhibit visible light semiconducting properties, allowing them to effectively absorb a significant portion of the solar spectrum.

### 3.2. Photocatalytic Hydrogen Evolution Study

Owing to their intriguing optoelectronic properties and unique mesoporous nano-architecture, the present mesoporous NC-based materials are expected to promote photocatalytic reactions. For this purpose, the photocatalytic performance of the synthesized materials was evaluated for the H_2_ evolution reaction from water. The effect of ZnS content on the photocatalytic H_2_ evolution activity of the nanocomposites was investigated under visible light irradiation (λ ≥ 420 nm), using 20 mg of catalyst dispersed in an aqueous solution (20 mL) containing 0.35 M Na_2_S and 0.25 M Na_2_SO_3_ as sacrificial reagents. Figure 5a compares the rates of H_2_ evolution (averaged over 3-h reaction period) obtained for the pure CdS and ZnS NCAs, along with the different ZnS/CdS catalysts. It is evident that the pure ZnS NCAs demonstrate nearly negligible H_2_ production activity (~4 μmol h^−1^), which is attributed to the insufficient visible light absorption due to the wide band gap of ZnS NCs (E_g_ = 3.85 eV). Additionally, despite the visible light response of the CdS NCAs (E_g_ = 2.51 eV), their H_2_ production performance was relatively poor (~30 μmol h^−1^), primarily due to the rapid recombination of photo-excited electron-hole pairs. Conversely, ZnS and CdS hybridization has a significant impact on the photocatalytic performance, markedly improving the H_2_ evolution rate. In particular, the photo-activity of ZnS/CdS NCAs was remarkably enhanced with increasing the loading amount of ZnS NCs, reaching a maximum H_2_-evolution rate of ~0.32 mmol h^−1^ at 50 wt.% ZnS content (50-ZnS/CdS catalyst). This activity is about 10.7 and 80 times higher than that of single CdS and ZnS NCAs, respectively (Figure 5a). Further increase in the ZnS content to 70 wt.% leads to a small decrease in the H_2_ evolution activity (0.27 mmol h^−1^ for the 70-ZnS/CdS catalyst), while the H_2_ evolution rate of the 90 wt.% ZnS-loaded sample (90-ZnS/CdS) was significantly reduced to 0.1 mmol h^−1^. As we will discuss in more detail later, the noticeably lower catalytic activities observed in the ZnS/CdS samples with low (10 wt.%) and high (90 wt.%) ZnS content can be attributed to the poorer charge transfer and separation ability, likely due to inadequate interfacial connectivity between unevenly distributed ZnS and CdS NCs in these assembled structures. The above findings thus indicate that the even (~1:1) mass distribution of ZnS and CdS NCs in the assembled structures plays a prominent role in the H_2_-evolution performance of the resulting materials. Consequently, we focused our further catalytic studies on reactions with the 50-ZnS/CdS catalyst. Note that blank experiments conducted in the absence of light or catalyst did not exhibit any evolution of H_2_, indicating that the observed production of H_2_ is exclusively attributed to photocatalytic reactions.

In addition to chemical composition, morphological effects may also affect the catalytic performance of ZnS/CdS catalysts. To assess this, we prepared a reference ZnS/CdS bulk catalyst with 50 wt.% ZnS content by physically mixing equal amounts of pre-formed ZnS and CdS microparticles (see experimental section). XRD and EDS characterization results indicated that the bulk reference catalyst consists of a polycrystalline mixture of ZnS and CdS (Appendix A) with 50 wt.% ZnS composition (Appendix A). Additionally, compared to the mesoporous 50-ZnS/CdS nanocomposite, the bulk ZnS/CdS microparticles possess a significantly lower surface area and pore volume (S_BET_ = 75 m^2^ g^−1^, V_p_ = 0.15 cm^3^ g^−1^) and a smaller energy band gap (E_g_ = 2.32 eV), as determined via N_2_ physisorption and UV–vis/NIR DRS measurements (see Appendix A and Table 1). Interestingly, the reference ZnS/CdS bulk sample demonstrated a H_2_ evolution rate of only ~30 μmol h^−1^, which is comparable to that of pure CdS NCAs and considerably lower than that of the mesoporous 50-ZnS/CdS nanocomposite (see Figure 5a). In addition, to better highlight the positive impact of the 3D mesoporous NC network structure on catalytic activity, we also prepared an untemplated reference nanocomposite, ZnS/CdS RNAs (RNAs: random NC-aggregates) containing 50 wt.% ZnS and compared its photocatalytic performance with that of the polymer-templated 50-ZnS/CdS catalyst. ZnS/CdS RNAs was synthesized following a template-free oxidative coupling of ZnS and CdS NCs, where the evaporation of solvent in the absence of polymer template resulted in the formation of close-packed NC-aggregates. EDS analysis indicated a ZnS content of 49.82 wt.% (Appendix A), which closely aligns with that of the 50-ZnS/CdS sample. However, in contrast to the polymer-templated nanocomposite, the ZnS/CdS RNAs exhibited a different isotherm profile, resembling a combination of type-I and type-IV with an H3 hysteresis loop, indicative of a dense microporous structure (S_BET_ ~114 m^2^ g^−1^, V_p_ = 0.07 cm^3^ g^−1^) with small interstitial voids (~1.5 nm) (see Appendix A). Under the same reaction conditions (i.e., 20 mg of catalyst dispersed in a 0.35 M Na_2_S and 0.25 M Na_2_SO_3_ solution), the ZnS/CdS RNAs catalyst yielded a visible light H_2_-evolution rate of ~0.21 mmol h^−1^ (see Figure 5a), which is approximately 1.5 times lower than that of the mesoporous 50-ZnS/CdS nanocomposite. Therefore, based on the above results, it can be deduced that the 3D open-pore morphology with high surface area and the NC-network structure with nanoscale interfaces are contributing factors that enhance mass transport phenomena and the number of accessible active sites and thus play a crucial role in boosting the H_2_ production performance of our mesoporous ZnS/CdS nanocomposite catalysts.

To further optimize the reaction conditions, we investigated the effects of different sacrificial electron donors and mass loadings of catalyst on photocatalytic H_2_ production performance. To this end, we examined a series of customary sacrificial reagents, including tertiary amines, such as triethanolamine (TEOA) and triethylamine (TEA), lactic acid, methanol and a Na_2_S/Na_2_SO_3_ mixture. The experiments were conducted in 20 mL of DI water, using a fixed amount (20 mg) of the mesoporous 50-ZnS/CdS catalyst. As depicted in Appendix A, the most favorable results were achieved when using a mixture of Na_2_S and Na_2_SO_3_ as sacrificial reagents. It is generally recognized that the utilization of S^2−^/SO_3_^2−^ species as electron donor serves as an effective strategy to enhance the H_2_-evolution photoactivity of metal-sulfide catalysts. The S^2−^/SO_3_^2−^ pairs not only consume the surface-reaching holes but also mitigate the anodic photocorrosion of sulfur-containing catalysts (such as CdS and ZnS) by replenishing the loss of lattice sulfur with S^2−^ ions from the reaction solution [9,15,53]. On this basis, the concentration of S^2−^/SO_3_^2−^ species during the photocatalytic reaction also plays an important role, as demonstrated by the control experiments performed using different concentrations of the Na_2_S/Na_2_SO_3_ mixture (Figure 5b). The results indicate that the H_2_-evolution activity of 50-ZnS/CdS scales almost linearly with the Na_2_S/Na_2_SO_3_ concentration and reaches a maximum of 0.58 mmol h^−1^ in a saturated aqueous electrolyte containing 1.4 M Na_2_S and 1.0 M Na_2_SO_3_. Furthermore, photocatalytic tests using different mass loadings of the 50-ZnS/CdS catalyst (dispersed in a 1.4 M Na_2_S and 1.0 M Na_2_SO_3_ aqueous solution) indicated that the rate of H_2_ evolution increased with the catalyst concentration, reaching a maximum at 1 g L^−1^ (Figure 5c). This result can be attributed to the saturation of light absorption by the catalyst particles. A further increase of the catalyst concentration (to 1.5 and 2 g L^−1^) results in a gradual reduction in H_2_ production performance (0.42 and 0.37 mmol h^−1^, respectively), likely due to the light-scattering and shielding effects of the excessive number of catalyst particles in the reaction mixture.

Under the optimized reaction conditions (i.e., 1 g L^−1^ catalyst concentration and 1.4 M Na_2_S and 1.0 M Na_2_SO_3_ sacrificial reagents), the 50-ZnS/CdS catalyst also demonstrated excellent photochemical stability. The stability was evaluated by conducting four consecutive 6-h photocatalytic H_2_-evolution runs under visible light irradiation (details shown in Section 2.6). As inferred from the results presented in Figure 5d, the 50-ZnS/CdS catalyst maintains its high H_2_-evolution activity without any signs of decline (within 5% experimental error) even after 24 h of irradiation. After stability tests, we performed EDS, XRD, XPS and N_2_ physisorption characterizations to investigate the chemical and structural stability of the reused catalyst. EDS analysis of the retrieved catalyst indicated a Cd/Zn atomic ratio of 15.62:23.42, which corresponds to a ZnS content of 50.29 wt.% (see Appendix A), very close to the initial ZnS loading amount of the fresh sample (50.37 wt.%). Additionally, no changes were observed in the crystal structure and chemical state of elements in the reused catalyst, as evidenced by its XRD pattern (Appendix A) and XPS spectra (Appendix A). The obtained results closely resembled those of the as-prepared 50-ZnS/CdS NCAs, substantiating the excellent chemical and structural stability of the catalyst during the examined conditions. However, N_2_ physisorption measurements on the reused sample revealed a relatively lower BET surface area (~200 m^2^ g^−1^) and pore volume (~0.26 cm^3^ g^−1^) compared to the fresh material (S_BET_ = 266 m^2^ g^−1^, V_p_ = 0.31 cm^3^ g^−1^) (Appendix A). In addition, pore size analysis indicated a slightly broader pore-size distribution, with a peak maximum at ~6.2 nm. These small changes in the textural parameters are probably related to some minor photocorrosion, which leads to a rearrangement of nanoparticles within the porous structure. Regardless, the mesoporosity and surface area of 50-ZnS/CdS NCAs still remain high enough after the prolonged catalytic use. The 4-times reused catalyst also demonstrated a persistent H_2_-evolution activity for two additional photocatalytic cycles (see Appendix A), providing further confirmation of its impressive photocatalytic stability and reusability under the examined conditions. Remarkably, over the 36-h irradiation period, an average H_2_-evolution rate of 0.58 mmol h^−1^ (or ~29 mmol g^−1^ h^−1^ mass activity) was achieved, which is associated with an apparent quantum yield (AQY) of 60% at λ = 420 nm, assuming that all incident photons are absorbed by the catalyst. To the best of our knowledge, this efficiency is among the highest reported thus far for CdZnS-based photocatalytic systems, according to previous studies [16,17,18,19,20,21,22,23,24,25,54,55,56,57,58]. A comparison of photocatalytic H_2_-evolution activities of some high-performance CdZnS-based photocatalysts is presented in Appendix A. Furthermore, to assess the influence of S^2−^/SO_3_^2−^ concentration on the photochemical stability of 50-ZnS/CdS, we also conducted photocatalytic H_2_-evolution recycling tests (i.e., successive 4-h photocatalytic runs) using the typical 0.35 M Na_2_S and 0.25 M Na_2_SO_3_ concentrations of sacrificial reagents. As shown in Appendix A, the lower S^2−^/SO_3_^2−^ concentration not only yielded an overall lower H_2_-evolution activity but also failed to prevent catalyst photocorrosion, as inferred from the gradual decline in H_2_-evolution rates observed during the second and third catalytic runs. Hence, this observation underscores the profound influence of the S^2−^/SO_3_^2−^ concentration on both the photoactivity and stability of the prepared mesoporous nanocomposites. Due to the large surface area and abundance of exposed active sites, the lower S^2−^/SO_3_^2−^ concentration is inadequate for the efficient consumption of surface-reaching holes, thus leading to lower rates of H_2_-evolution and gradual photocorrosion of the nanocomposites.

### 3.3. Photocatalytic Mechanism

To clarify our understanding of the interfacial electronic interactions between the CdS and ZnS components and gain insight into the photocatalytic H_2_ production mechanism of the CdS/ZnS nanocomposites, we performed electrochemical impedance spectroscopy (EIS) and photoluminescence (PL) measurements. Figure 6a displays the Mott–Schottky (M–S) plots, i.e., the inverse square capacitance (1/C^2^) versus applied potential diagrams, for the mesoporous CdS, ZnS and ZnS/CdS NCAs. The M–S plot of the bulk ZnS/CdS material with 50 wt.% ZnS is shown in Appendix A. All the samples showed positive slopes in the M–S plots, indicating n-type conductivity. Furthermore, by extrapolating the linear fits of the M–S curves to 1/C^2^ = 0, we determined the flat band potential (E_FB_) of all the materials, and the results are compared in Table 2. All the measured potentials are reported relative to the reversible hydrogen electrode (RHE) at pH 7. For single-component CdS and ZnS NCAs, the estimated E_FB_ levels were −0.70 V and −1.12 V, respectively, while the E_FB_ positions of the ZnS/CdS materials range from −0.74 V to −1.06 V. Generally, E_FB_ represents the electrochemical Fermi level energy (E_F_) of the materials under the examined conditions, which is approximately 0.1–0.3 eV below the conduction band (CB) edge in highly doped n-type semiconductors [59]. In fact, the location of the CB edge potential (E_CB_) can be calculated from E_FB_ using Equations (4) and (5), presuming that the donor dopants are completely ionized [60].
(4)$$N_{D}=N_{C}\cdot e^{\left[-\left(E_{\mathrm{CB}}-E_{F}\right)/\mathrm{kT}\right]}$$
where E_F_ is the Fermi level energy, N_D_ is the electron donor density and N_c_ is the density of the effective states in the CB, which is given by:(5)$$N_{c}=2\cdot {\left(\frac{2\pi \cdot m_{e}^{*}\cdot k\cdot T}{h}\right)}^{3/2}$$
where h is the Planck constant, k is the Boltzmann constant, T is the temperature in Kelvin and $m_{e}^{*}$ is the effective mass of electrons in the CB.

In Equation (5), assuming that m_e_* ≈ m_o_, in which m_o_ is the mass of a free electron, N_c,_ was estimated to be ~2.5 × 10^19^ cm^−3^. In addition, the donor density (N_D_) of the studied catalysts was determined from the slope of the Mott–Schottky linear fits (see Table 2). The resultant N_D_ values presented a progressive increase from 2.02 × 10^16^ to 4.90 × 10^16^ cm^−3^ with an increase in the ZnS ratio from 0 to 50 wt.%, followed by a relative decrease (to 2.32 × 10^16^ and 3.32 × 10^16^ cm^−3^) for the samples with higher ZnS contents (70 and 90 wt.%). Therefore, based on the obtained E_FB_ and N_D_ results, the E_CB_ levels of the prepared materials were calculated, and the valence band (VB) edge potentials (E_VB_) were subsequently determined by adding the respective optical band gaps (E_g_) obtained from UV-vis/NIR spectra. Table 2 summarizes all the calculated band-edge potentials, from which a representative band-edge diagram for each catalyst was constructed, as illustrated in Figure 6b. By analyzing the band diagrams in conjunction with the corresponding electrochemical results, several important conclusions can be drawn. Firstly, the CB edge of all the prepared catalysts is positioned well above the H_2_O/H_2_ reduction potential (−0.41 V vs. RHE at pH 7), demonstrating the thermodynamic ability of these materials to drive water reduction. Moreover, a clear cathodic shift is observed in the CB edge of the ZnS/CdS nanocomposites with increasing ZnS content (from 10 to 90 wt.%). This shift can be explained by the higher E_FB_ level of ZnS (−1.12 V) relative to that of CdS (−0.7 V), which creates a built-in potential difference after their contact. As a result, an electron flow from ZnS to CdS will take place at the ZnS/CdS contact interface until thermodynamic equilibrium is reached between the two materials. Consequently, upon enrichment with ZnS NCs, the accumulation of electrons in CdS will progressively shift the CB edge level of the nanocomposites in a more negative (cathodic) direction, positioning it between those of pure ZnS and CdS, as evidenced by the obtained EIS results. The electron transfer from ZnS to CdS in the nanocomposites is also supported by the XPS results as well as the increasing N_D_ values with the loading amount of ZnS. The relatively lower N_D_ values obtained for the nanocomposites with high ZnS content (70 and 90 wt.%) and the ZnS/CdS bulk reference (see Table 2) likely suggest a weaker charge transfer effect between the ZnS and CdS components in these materials. To further elucidate this, the charge-transfer properties of the prepared catalysts were investigated via Nyquist plot analysis. EIS Nyquist measurements were conducted in 0.5 M Na_2_SO_4_ electrolyte over a frequency span of 1 Hz–100 kHz using an AC amplitude of 10 mV and a DC bias of −1.3 V (vs. Ag/AgCl, saturated KCl). Figure 6c compares the Nyquist plots of pure CdS and ZnS NCAs with those of the ZnS/CdS nanocomposites with different ZnS content. The corresponding Nyquist plot of the ZnS/CdS bulk analogue is shown in Appendix A. The charge-transfer resistance (R_ct_) of each sample was obtained by fitting the Nyquist plots to a Randles equivalent circuit model (inset of Figure 6c and Appendix A), and the results are listed in Table 2 and Appendix A. In brief, the simulated results indicate that all the ZnS/CdS nanocomposites possess lower R_ct_ values (~285–747 Ω) than the pure CdS (~1312 Ω) and ZnS (~1362 Ω) NCAs. This suggests more favorable charge-transfer dynamics at the ZnS/CdS heterojunctions, which contribute to their enhanced photocatalytic performance. Among them, the 50-ZnS/CdS catalyst exhibits the lowest R_ct_ (285 Ω), which aligns with its superior charge transfer and photoreduction activity. Comparatively, the bulk reference catalyst (ZnS/CdS bulk) features a significantly higher R_ct_ (1499 Ω), despite having a similar composition and band-edge structure to the 50-ZnS/CdS nanocomposite. Collectively, these findings suggest that, in addition to the chemical composition, the nanoscale dimensionality of the ZnS/CdS contact interfaces plays a prominent role in enhancing the charge transfer dynamics and achieving superior photocatalytic efficiency.

Additional evidence for the importance of ZnS/CdS nanoscale heterojunctions in governing the efficiency of charge carrier transfer and separation processes is provided by time-resolved photoluminescence (TR-PL) measurements. Figure 6d depicts the TR-PL emission spectra of mesoporous CdS and 50-ZnS/CdS NCAs along with that of the ZnS/CdS bulk material, recorded using 375 nm pulse laser excitation. To obtain the carrier lifetimes, the PL decay curves were fitted with a biexponential function: F(t) = y_0_ + Σ_i_A_i_ e^−t/τi^ (i = 1, 2), where A_i_ is the amplitude fraction (Σ_i_A_i_ = 1) and τ_i_ is the carrier lifetime of each component. The two time-components in the equation reflect the fast radiative charge-carrier recombination at the surface (τ_1_) and slow recombination of excitons in the bulk (τ_2_), respectively. The estimated fast and slow lifetimes of carriers are tabulated in Appendix A, from which the average lifetimes (τ_av_) were calculated to be approximately 3.20 ± 0.017 ns and 3.70 ± 0.025 ns for the mesoporous CdS and 50-ZnS/CdS NCAs, respectively and 3.16 ± 0.011 ns for the ZnS/CdS bulk catalyst. These results attest to an improved interfacial charge transfer and separation process within the ZnS/CdS NC-assembled structure and thereby to a more efficient utilization of photo-excited electrons for H_2_ evolution. Moreover, as shown in Appendix A, 50-ZnS/CdS has a lower percentage (65.0%) of τ_1_-carriers than pure CdS (72.5%), indicating that the surface recombination of electron-hole pairs over CdS was inhibited upon hybridization with ZnS NCs. The efficient charge separation in the 50-ZnS/CdS catalyst was also confirmed via steady-state PL emission spectra obtained via excitation at 380 nm. As depicted in Appendix A, the PL spectrum of pure CdS NCAs exhibits a strong emission at ~494 nm (2.51 eV) corresponding to the band-edge excitonic relaxation of CdS NCs. Comparatively, the PL intensity of this peak is almost vanished in the 50-ZnS/CdS sample, indicating that the majority of photogenerated electron-hole pairs are dissociated over the ZnS/CdS structure, thus suppressing their recombination rate. However, this condition implies the spatial separation of charge carriers across the ZnS/CdS junctions, which is not supported by the apparent band-edge alignment between ZnS and CdS in the nanocomposites. In particular, as inferred from the band diagram of Figure 6b, the junction between ZnS and CdS is expected to form a type-I (straddling gap) band-alignment, which cannot promote the spatial separation of charge-carriers. Instead, the type-I junction configuration favours the accumulation of electrons and holes in CdS, thus increasing their recombination probability. Therefore, a more complex charge-dissociation mechanism likely occurs at the ZnS/CdS junctions, potentially involving interfacial charge transfer pathways between the band-edges of CdS and defect-induced energy levels commonly found in ZnS. This assumption is supported by the PL spectrum of ZnS NCAs (Appendix A), which exhibits not only an emission peak at 322 nm (3.85 eV) due to the band-edge relaxation but also multiple emission bands with peaks at 398 nm (3.12 eV), 432 nm (2.87 eV), 473 nm (2.62 nm) and 562 nm (2.21 eV) that are attributed to the radiative recombination of excitons through defect states. Typically, four types of point defects can be present in ZnS particles: sulfur vacancies (V_S_), zinc vacancies (V_Zn_), interstitial zinc atoms (I_Zn_) and interstitial sulfur atoms (I_S_). These defects introduce energy states within the band gap of ZnS, with the V_S_ and I_Zn_ levels located bellow the CB (donor states), while V_Zn_ and I_S_ create energy levels above the VB (acceptor states) [61]. It has been reported that the energy levels of vacancies are deeper than those of interstitial states, with I_Zn_ situated closer to the CB than V_S_, and I_S_ closer to the VB than V_Zn_ [62]. In this context, the PL emission bands centered at 3.12 eV and 2.87 eV can be attributed to electron transitions from the I_Zn_ and V_S_ states to the VB of ZnS, while the bands around 2.62 eV and 2.21 eV are likely associated with excitonic relaxations from V_S_ to I_S_ and from I_Zn_ to V_Zn_, respectively, as illustrated in Figure 7a [63].

On the basis of the above results, a plausible charge transfer and photocatalytic H_2_ production mechanism is proposed for the ZnS/CdS nanocomposite catalysts (Figure 7b). As revealed by the EIS analysis, CdS has a lower E_FB_ level than ZnS and hence, upon hybridization, electrons tend to flow from ZnS to CdS until their electrochemical potentials reach equilibrium. During this process, the loss of electrons from ZnS will result to an upward band bending near the interface (depletion layer), while the simultaneous increase in electron density at the CdS side will lead to a downward bending of the energy bands (accumulation layer). We found that, at optimum ZnS composition (50 wt.%), a built-in potential of about 0.4 V (as determined from Mott–Schottky results) will be established at the ZnS/CdS interfaces, which consequently creates a strong internal electric field with a direction from ZnS to CdS. Upon visible light irradiation, CdS undergoes excitation and generates electron and holes in the CB and VB, respectively. On the other hand, as inferred via PL measurements (see Figure 7a), electrons in ZnS can only be excited to the interband V_S_ and I_Zn_ states through single or multiple visible light-photon absorption, leaving behind holes in the VB, as well as the V_Zn_ and I_S_ acceptor states. Due to the built-in electric field at the ZnS/CdS interfaces, the photoexcited electrons in the CB of CdS will drift towards ZnS and be captured at the V_S_ and I_Zn_ donor states, together with the photogenerated electrons in ZnS. Such electron transfer route is supported by the direction of the internal electric field and the fact that the energy levels of V_S_ and I_Zn_ donor states (ranging from −0.3 to −0.6 V) are located just below the E_CB_ of CdS (−0.7 V), as illustrated in Figure 7b. On the other hand, the holes in the VB, V_Zn_ and I_S_ states of ZnS are driven by the electric field to the VB of CdS, where they accumulate alongside the holes in CdS. This hole transport pathway from ZnS to CdS is not only influenced by the internal electric field but is also thermodynamically feasible, as the E_VB_ of ZnS (2.56 V) as well as the energy levels of V_Zn_ and I_S_ acceptor states (~1.65–2.31 V) are located below the E_VB_ of CdS (1.62 V). Additional indication for the accumulation of holes in CdS was obtained through EDS analysis on the 50-ZnS/CdS catalyst retrieved after the control photocatalytic experiment in which TEOA was used as sacrificial electron donor. Regarding this, because TEOA exhibits a slower scavenging ability for the photogenerated holes compared to the S^2−^/SO_3_^2−^ species (see Appendix A), it is not expected to efficiently prevent the anodic photocorrosion of CdS by the accumulated photoholes. Indeed, as shown in Appendix A, the EDS results of the retrieved catalyst (50-ZnS/CdS-c) revealed a Zn/Cd atomic ratio of ~1.82, representing an increase of approximately 17.6% compared to the fresh sample (Zn/Cd ≈ 1.50). Additionally, the ZnS content increased from 50.34 to 55.14 wt.% after the photocatalytic reaction. Therefore, these results explicitly demonstrate a significant loss of Cd due to the photocorrosion of CdS, further proving its role as the hole acceptor in the ZnS/CdS heterojunctions. Consequently, via this defect-induced quasi-type-II charge transfer scheme, the photogenerated charge carriers in CdS and ZnS can be efficiently separated at the ZnS/CdS nanojunctions. The proposed charge transfer and separation mechanism endows electrons trapped in the donor states of ZnS with prolonged ability to participate in the catalytic reduction of H_2_O to generate H_2_. At the same time, the holes gathered in the VB of CdS can effectively oxidize the sacrificial S^2−^/SO_3_^2−^ species, thus increasing the photochemical activity and stability of the catalyst. Such photocatalytic mechanism is consistent with the results from the above electrochemical and spectroscopic studies. In addition, the mesoporous ZnS/CdS NCAs, owing to the 3D open-pore structure and high internal surface area, provide a high density of surface-active sites for redox reactions. As thus, the overall photocatalytic performance of these catalysts is significantly improved, highlighting their potential for efficient solar-fuel production.

## 4. Conclusions

In summary, mesoporous dual-semiconductor ZnS/CdS nanocomposites have been successfully synthesized via a cross-linking polymerization of 5–7 nm-sized CdS and ZnS nanoparticles in the presence of a block–copolymer template. The ZnS/CdS NCAs are composed of tightly interconnected ZnS and CdS NCs that form a 3D open-pore structure with large internal surface area (circa 243–285 m^2^ g^−1^) and narrow pore size distribution (circa 5.8–6.3 nm), as confirmed by elemental microprobe analysis, X-ray diffraction, electron microscopy, X-ray photoelectron spectroscopy and N_2_ porosimetry measurements. By carefully adjusting the ZnS composition in the ZnS/CdS nanocomposites, we showed that the interfacial electronic connectivity between the ZnS and CdS components can be finely tuned to achieve enhanced charge-carrier separation efficiency. Valuable insights into the charge transfer pathways in the ZnS/CdS heterojunctions were provided by electrochemical impedance spectroscopy and photoluminescence studies. The results corroborate to a quasi-type-II interfacial charge-transfer scheme between the defect states in ZnS and band-edges of CdS, that promote the charge transfer and separation for efficient H_2_ production. As a result, the optimized mesoporous ZnS/CdS catalyst with 50 wt.% ZnS content achieved an impressive photon-to-hydrogen conversion efficiency of about 60% at 420 nm, which ranks among the highest reported values compared to other CdZnS-based photocatalytic systems. Moreover, prolonged photocatalytic recycling tests demonstrated the catalyst’s exceptional stability for H_2_ evolution. Overall, this study not only enhances our understanding of the intricate charge transfer dynamics at the nanoscale ZnS/CdS junctions but also underscores the promising potential of these mesoporous nano-architectures to serve as efficient photocatalysts for solar-to-hydrogen conversion technologies.

## Figures and Tables

**Figure 1 nanomaterials-13-02426-f001:**
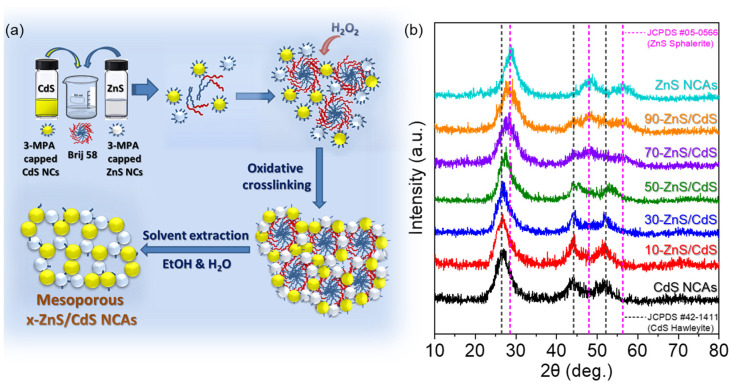
(**a**) Schematic representation of the synthetic procedure of mesoporous ZnS/CdS NCAs. (**b**) Typical XRD patterns of mesoporous CdS and ZnS NCAs and the as-prepared ZnS/CdS nanocomposites with different ZnS loadings.

**Figure 2 nanomaterials-13-02426-f002:**
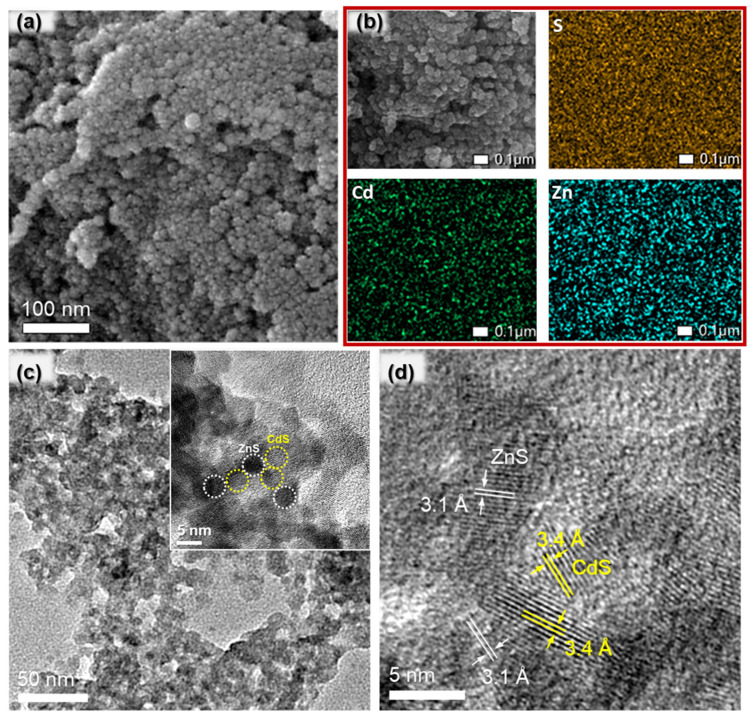
(**a**) Typical FE-SEM image, (**b**) EDS mappings of Cd, Zn and S elements, (**c**) TEM image and high magnification TEM image (inset) showing CdS (yellow circles) and ZnS (white circles) nanoparticles in intimate contact, and (**d**) HRTEM image for the mesoporous 50-ZnS/CdS nanocomposite. In HRTEM, the lattice fringes with d-spacings of 3.1 and 3.4 Å are indexed to the (111) planes of the cubic structures of ZnS (white lines) and CdS (yellow lines), respectively.

**Figure 3 nanomaterials-13-02426-f003:**
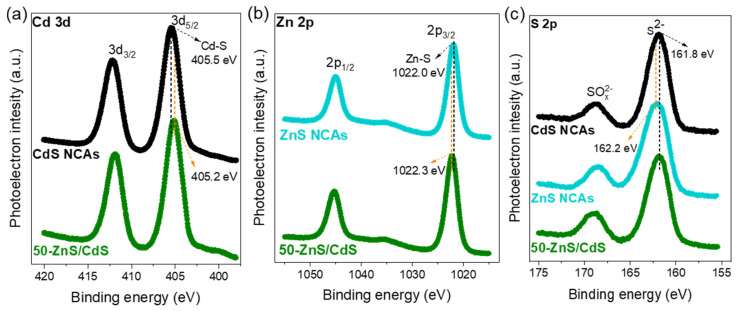
(**a**) Cd 3d, (**b**) Zn 2p and (**c**) S 2p XPS core-level spectra of mesoporous CdS and ZnS NCAs and the 50-ZnS/CdS nanocomposite.

**Figure 4 nanomaterials-13-02426-f004:**
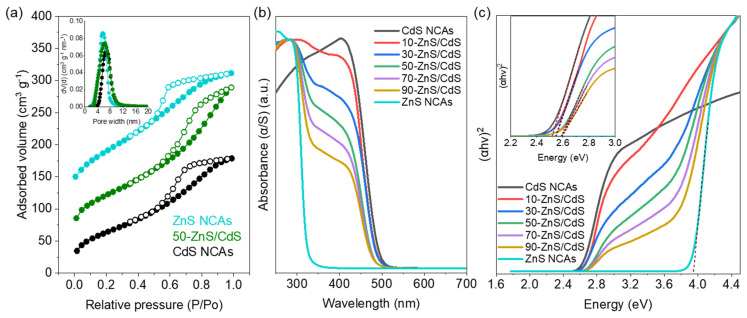
(**a**) Comparative N_2_ adsorption (filled cycles) and desorption (open cycles) isotherms at −196 °C and (inset) the corresponding NLDFT pore size distribution plots calculated from the adsorption branch of isotherms for the mesoporous CdS, ZnS and 50-ZnS/CdS NCAs. For clarity, the isotherms of 50-ZnS/CdS and ZnS NCAs are offset by 50 and 100 cm^3^ g^−1^, respectively. (**b**) UV-vis/NIR diffuse reflectance spectra and (**c**) Tauc plots (i.e., the curves of (αhν)^2^ versus photon energy (hν), where α, h and ν are the absorption coefficient, Planck’s constant and light frequency, respectively) of mesoporous CdS, ZnS and ZnS/CdS NCAs with different wt.% ZnS content. Inset: magnification of the Tauc plot in the energy range of 2.2–3.0 eV for clarity.

**Figure 5 nanomaterials-13-02426-f005:**
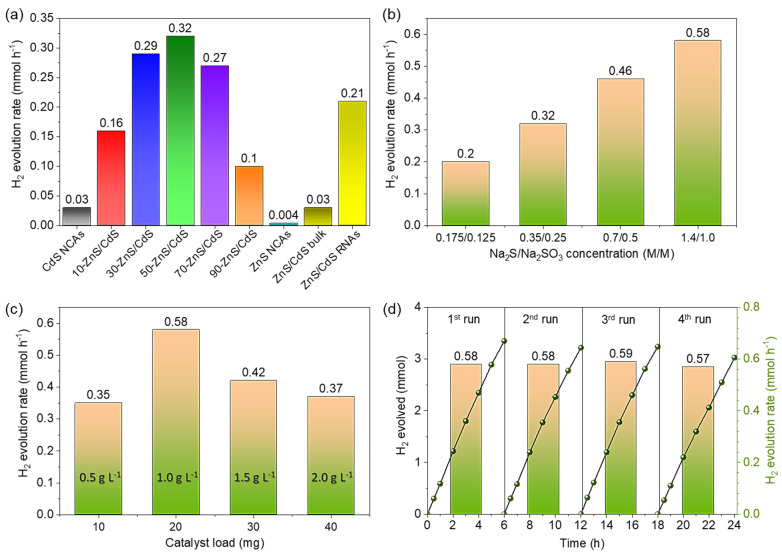
(**a**) Photocatalytic H_2_ evolution activities for mesoporous CdS, ZnS and the different ZnS/CdS NCAs, together with the ZnS/CdS bulk and ZnS/CdS RNAs reference catalysts with 50 wt.% ZnS content. The photocatalytic reactions were carried out in an airtight reactor, using 1 mg mL^−1^ catalyst concentration in a 0.35 M Na_2_S and 0.25 M Na_2_SO_3_ aqueous electrolyte. (**b**) Photocatalytic H_2_ evolution rates for the mesoporous 50-ZnS/CdS catalyst (1 mg mL^−1^ catalyst concentration) using different concentrations of Na_2_S/Na_2_SO_3_ reagents and (**c**) different mass loadings of the catalyst in a 1.4 M Na_2_S and 1.0 M Na_2_SO_3_ aqueous electrolyte. (**d**) Photocatalytic recycling tests of the 50-ZnS/CdS catalyst (1 mg mL^−1^) in a 1.4 M Na_2_S and 1.0 M Na_2_SO_3_ aqueous electrolyte. All the H_2_-evolution rates obtained as an average over the initial 3-h reaction period. All photocatalytic tests were conducted under visible light irradiation using a 300 W Xenon light source with a UV cutoff filter (λ ≥ 420 nm).

**Figure 6 nanomaterials-13-02426-f006:**
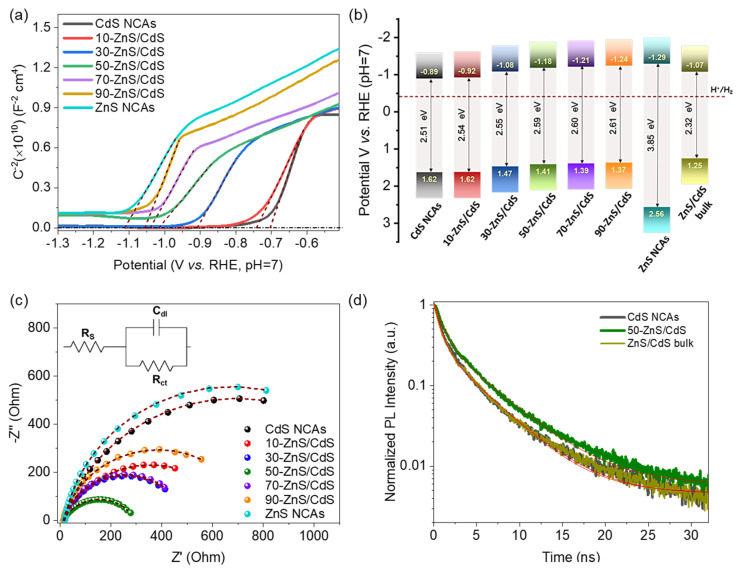
(**a**) Mott–Schottky plots where the E_FB_ values are obtained from the intercepts of the extrapolated linear fits of the 1/C^2^ vs. potential curves, (**b**) energy band diagrams (the band-edge diagram of the ZnS/CdS bulk sample with 50 wt.% ZnS is also given) and (**c**) EIS Nyquist plots and equivalent circuit model (inset) for the mesoporous CdS, ZnS and ZnS/CdS NCAs. (**d**) Comparative time-resolved photoluminescence (TR-PL) decay curves of mesoporous CdS and 50-ZnS/CdS NCAs and ZnS/CdS bulk reference with 50 wt.% ZnS content.

**Figure 7 nanomaterials-13-02426-f007:**
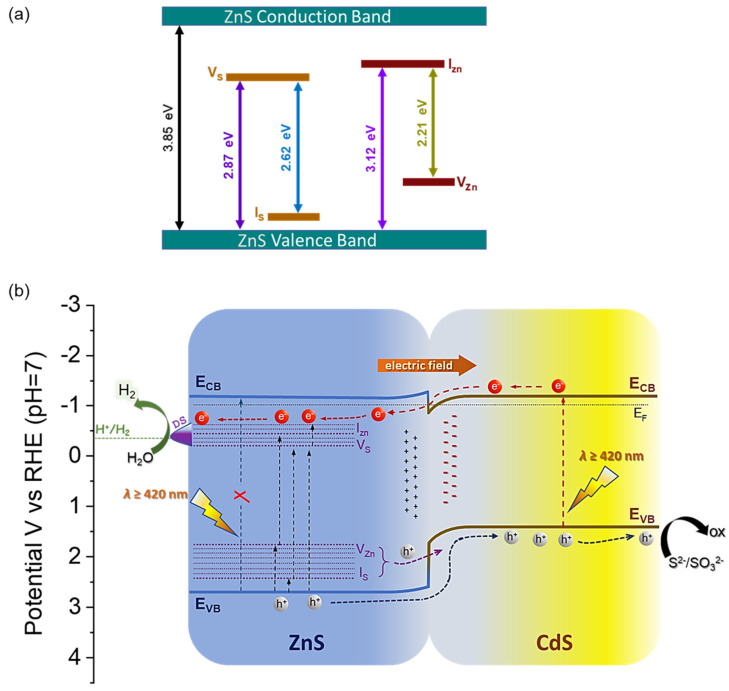
(**a**) Schematic energy diagram for the point defect states in ZnS (V_S_ = sulfur vacancy, I_S_ = interstitial sulfur, V_Zn_ = zinc vacancy, and I_Zn_ = interstitial zinc). The defect-state energy levels are estimated according to the peak positions of the ZnS NCAs PL spectrum. (**b**) Schematic illustration of the proposed quasi-type-II charge-transfer and visible light photocatalytic H_2_-generation mechanism for the ZnS/CdS nanocomposites. The presented energy levels correspond to the band diagram of the 50-ZnS/CdS catalyst (E_CB_ = conduction band level, E_VB_ = valence band level, E_F_ = Fermi level and DS = donor states).

**Table 1 nanomaterials-13-02426-t001:** Textural parameters and energy band gap values for the mesoporous CdS, ZnS and ZnS/CdS NCAs. The respective data for the physical mixture of bulk ZnS and CdS microparticles with 50 wt.% ZnS (ZnS/CdS bulk) are also shown, determined from the N_2_ physisorption and optical absorption measurements presented in Appendix A.

Sample	BET Surface S_BET_ (m^2^ g^−1^)	Pore Volume V_p_ (cm^3^ g^−1^)	Pore Size D_p_ (nm)	Energy Gap ^1^ E_g_ (eV)
CdS NCAs	234	0.28	6.8	2.51 ± 0.01
10-ZnS/CdS	243	0.29	6.3	2.54 ± 0.02
30-ZnS/CdS	252	0.30	6.3	2.55 ± 0.01
50-ZnS/CdS	266	0.31	6.0	2.59 ± 0.01
70-ZnS/CdS	277	0.31	5.9	2.60 ± 0.01
90-ZnS/CdS	285	0.32	5.8	2.61 ± 0.01
ZnS NCAs	334	0.33	5.8	3.85 ± 0.02
ZnS/CdS bulk	75	0.15	~7.8	2.32 ± 0.01

^1^ Energy band gaps determined from Tauc plots for direct allowed transitions. E_g_ values for all the prepared ZnS/CdS materials were estimated from the respective absorption bands in the visible light region of the Tauc plots, as shown in the insets of Figure 4 and Appendix A.

**Table 2 nanomaterials-13-02426-t002:** Electrochemical results obtained from EIS analyses for the mesoporous CdS, ZnS and ZnS/CdS NCAs and the ZnS/CdS bulk reference with 50 wt.% ZnS content.

Sample	E_FB_	E_CB_	E_VB_	Donor Density (N_D_) (cm^−3^)	Charge Transfer Resistance (R_ct_) (Ohm)
	**(V vs. RHE, pH = 7)**				
CdS NCAs	−0.70	−0.89	1.62	2.02 × 10^16^	1312
10-ZnS/CdS	−0.74	−0.92	1.62	2.86 × 10^16^	686
30-ZnS/CdS	−0.91	−1.08	1.47	3.35 × 10^16^	474
50-ZnS/CdS	−1.02	−1.18	1.41	4.90 × 10^16^	285
70-ZnS/CdS	−1.04	−1.21	1.39	3.32 × 10^16^	486
90-ZnS/CdS	−1.06	−1.24	1.37	2.32 × 10^16^	747
ZnS NCAs	−1.12	−1.29	2.56	3.38 × 10^16^	1362
ZnS/CdS bulk	−0.90	−1.07	1.25	2.99 × 10^16^	1499

## Data Availability

The data presented in this study are available in article and Appendix A.

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
