# Peer review of "Mesoporous Dual-Semiconductor ZnS/CdS Nanocomposites as Efficient Visible Light Photocatalysts for Hydrogen Generation"

_nanomaterials, 2023, doi:10.3390/nano13172426_

Round 1

Reviewer 1 Report

Comments:

In this manuscript, the authors have successfully designed and synthesized interlinked ZnS/CdS nanoparticle composites with a 3D open-pore structure. This unique structure enhances the exposure of active sites, leading to a promising catalytic performance. Overall, I recommend the publication of this study, after addressing the following questions:

1.     The authors cannot ignore the distribution and content of polymers in the porous ZnS/CdS nanocomposites, as well as the potential roles they may play in the photocatalytic process.

2.     The authors need to provide more detailed explanations regarding the key synthesis parameters used to obtain this interlinked 3D porous structure. For instance, how to prevent excessive aggregation of small particles, how to ensure the robustness of this 3D structure, and how to control the crystallinity of the small particles.

3.     I suggest the author provide additional explanations regarding the stability of this structure in water and its hydrophilic or hydrophobic characteristics. Particularly, in the context of photocatalysis, the stability of this structure in a water environment is crucial. The author can elaborate on how the structure's degradation or aggregation in water is prevented, and describe the methods used to ensure its long-term stability in aqueous conditions.

Reviewer 2 Report

The present work reported the synthesis of dual-semiconductor ZnS/CdS nanocomposites by cross-linking polymerization of CdS and ZnS nanoparticles using a block-copolymer template. The resultant nanocomposites showed a 3D open-pore structure with a large surface area (up to 285 m2/g) and uniform pores. The optimized one containing 50% wt. ZnS exhibited high and stable photocatalytic activity under visible light with an H2 evolution rate of 29 mmol g-1h-1 and an apparent quantum yield of 60% at 420 nm. This excellent activity results from the porous structure and efficient charge transfer through ZnS/CdS nanointerfaces. In this work, the authors thoroughly characterized as-prepared catalysts using different techniques, elucidating their structure and properties. They discussed in detail the photocatalytic operation of as-prepared catalysts from various results obtained from the characterizations. I recommend this work for publication on Nanomaterials after a careful revision.

Here are my several concerns:

1. The quality of the TEM image in Figure 2c is low to observe nanoparticles. A new image with higher quality should be provided.

2. XPS survey should be provided in Supplementary Materials.

3. The authors showed that using the electrolyte containing 1.4 M Na2S and 1.0 M Na2SO3 resulted in the highest activity, and they used this condition for several tests (Figures 5c and d). Meanwhile, the tests in Figure 5a and Figure S3 were performed at lower electrolyte concentrations of 0.35 M Na2S and 0.25 M Na2SO3. An identical reaction condition should be used throughout the work.

4. The activity comparison of as-prepared mesoporous nanocomposites with only ZnS/CdS bulk prepared by co-precipitation is insufficient to highlight the role of 3D mesoporous structure in promoting photoactivity. The authors should additionally provide photoactivity of nanocomposite photocatalysts formed by assembling ZnS and CdS nanoparticles without a block-copolymer template.

5. Stability is usually one of the main challenges for chalcogenide photocatalysts. In the work, the photoactivity of the optimized sample was found stable after 24 hours. However, a significant structure change with a shrunken surface area could be observed after this period. Therefore, the stability of the prepared photocatalysts should be verified for more extended periods (e.g., 36-48 hours).

6. The procedure of the stability test should be detailed in section 2.6.

7. Experimental reaction conditions for 50-ZnS/CdS AC 3 and 50-ZnS/CdS-c 4 samples in Table S3 should be detailed.

8. The concentrations of Na2S and Na2SO3 strongly affect the photoactivity of prepared photocatalysts. Do they affect the stability of prepared photocatalysts? This point should be verified since S2-/SO32- pairs have a critical role in mitigating the photocorrosion of photocatalysts by replenishing the loss of lattice sulfur.

Reviewer 3 Report

All in all a highly valuable paper containing a lot of interesting information on this new nanocomposite material. To raise the value of the paper even more I recommend to perform some checks on the data, especially visible-light H2 evolution activity / apparent quantum yield and Mott-Schottky analysis data.

My comments in detail:

Lines 92-102: are in principle a repetition of the abstract and an anticipation of the results; I recommend to omit these lines.

Remark on Lines 119-120: I am wondering on the addition of ammonia to adjust pH to 10; I think this would result in a precipitation of hydroxides?

Line 236, formula (3): to be correct you should note that you omitted the term with kT in the formula; also a reference for the formula should be added, see for instance: Mott–Schottky Analysis of Photoelectrodes: Sanity Checks Are Needed, Kevin Sivula, ACS Energy Letters 2021 6 (7), 2549-2551, DOI: 10.1021/acsenergylett.1c01245; in this paper includes also “Here are a few sanity checks that can be performed to quickly correct and assess the validity of your (Mott-Schottky analysis) results”

Lines 260-261: “oxidative polymerization of metal-sulfide NCs (via S-S bond formation) [35].” In the cited reference Raman measurements on interconnected CdSe have been used to verify the homonuclear bond formation. Did you also perform Raman or FTIR measurements on your samples to verify your hypothesis?

Line 291: how has the instrumental line broadening as correction been determined to evaluate the average domain size via Scherrer equation?

Lines 331-332: “attributed to some oxidized sulfur species (SOx2-) formed at the particle surface during the synthesis process and/or air handling of the samples” Figure 3c indicates that a quite high amount of sulfate species is present (20%?)!” Could the amount be quantified by cross section considerations? On the other side, the corresponding S 2p region of the 50-ZnS/CdS catalyst retrieved after 24-h of photocatalytic reaction in Figure S4 does not reveal any sulfate species at all; can you explain this observation?

Lines 388-390: “…mesoporous samples exhibit a significant blue-shift in Eg, which is reflective of the quantum confinement effect”: as significant changes can only be regarded in the case errors are specified; throughout the paper I am missing the errors, as for instance in Table S1 in the specification of the atomic contents according to (standardless!) EDS analysis; as data at several locations have been accumulated, errors would indicate local variations. Also the shifts in band gaps with 0.1 – 0.2 eV are rather small (compared to e.g. Hiemer et al., doi.org/10.1002/open.202200232). I also consider important to specify errors for the average lifetimes after the fit procedure in the section starting from line 617 ff. Please improve here the manuscript by specifying errors where possible.

Lines 411-412: in the table caption a reference to Figure S2 should be added. The energy gap values in Table 1 are remarkable with respect to the large change of Eg between 90_ZnS/CdS and ZnS NCAs from 2.61 to 3.85 eV. I recommend to add a short discussion on this point.

Lines 471-473: “The S2-/SO32- pairs not only consume the surface-reaching holes, but also mitigate the anodic photocorrosion of sulfur-containing catalysts (such as CdS and ZnS) by replenishing the loss of lattice sulfur with S2- ions from the reaction solution[9,15,45].”

First of all I am wondering here on the mixture of sulfide and sulfite, as the two compounds might react with another to form sulfur as precipitate (comproportionation). Did you check this?

Secondly, it is reported that the usage of artificial reagents may lead to an overdetermination of the H2 evolution activity as these compounds result in a production of hydrogen SH- + h = S +  ½ H2 ; especially this is a very striking feature in Figure 5(b); as reference for this point see for instance: “Do Sacrificial Donors Donate H2 in Photocatalysis?” Federica Costantino and Prashant V. Kamat, ACS Energy Letters 2022 7 (1), 242-246, DOI: 10.1021/acsenergylett.1c02487. A citation from this paper: “Simply ignoring the participation of a sacrificial donor in a photocatalysis experiment and making claims such as “high photon conversion efficiency” or “highly efficient photocatalyst” undermine the research advances.”

Please consider this paper and check your data carefully on hydrogen evolution from the sacrificial reagents. See also e.g. lines 529-532 and 742-744. This is reason why I qualify the manuscript as “major revisions needed”.

Minor corrections needed:

At some positions the greek symbols are replaced by normal letters: e.g. line 169 Al Ka; line 386: ahv and hv, also lines 404 and 405.

In Line 349 the number “2” is written as subscript but should read as H2.

In line 643 it should be Figure 6b.

Round 2

Reviewer 2 Report

The authors made a meticulous revision that addresses all my concerns. I recommend publication of this work in its present form. 

Reviewer 3 Report

Thankyou very much for the careful consideration of my comments and the changes made accordingly in the manuscript.